# A Molecular Hybrid of the GFP Chromophore and 2,2′-Bipyridine: An Accessible Sensor for Zn^2+^ Detection with Fluorescence Microscopy

**DOI:** 10.3390/ijms25063504

**Published:** 2024-03-20

**Authors:** Attila Csomos, Miklós Madarász, Gábor Turczel, Levente Cseri, Gergely Katona, Balázs Rózsa, Ervin Kovács, Zoltán Mucsi

**Affiliations:** 1Department of Chemistry, Femtonics Ltd., Tűzoltó utca 59, H-1094 Budapest, Hungary; attila.csomos@femtonics.eu (A.C.); zoltan.mucsi@uni-miskolc.hu (Z.M.); 2Hevesy György PhD School of Chemistry, Eötvös Loránd University, Pázmány Péter sétány 1/A, H-1117 Budapest, Hungary; 3BrainVisionCenter, Liliom utca 43, H-1094 Budapest, Hungary; mmadarasz@brainvisioncenter.com (M.M.); levente.cseri@brainvisioncenter.com (L.C.); 4NMR Research Laboratory, HUN-REN Research Centre for Natural Sciences, Magyar Tudósok körútja 2, H-1117 Budapest, Hungary; turczel.gabor@ttk.hu; 5Department of Organic Chemistry & Technology, Budapest University of Technology & Economics, 3. Muegyetem rakpart, H-1111 Budapest, Hungary; 6Two-Photon Measurement Technology Research Group, Pázmány Péter Catholic University, Práter utca 50/a, H-1083 Budapest, Hungary; 7Laboratory of 3D Functional Network and Dendritic Imaging, Institute of Experimental Medicine, Szigony utca 43, H-1083 Budapest, Hungary; 8Polymer Chemistry and Physics Research Group, HUN-REN Research Centre for Natural Sciences, Magyar Tudósok körútja 2, H-1117 Budapest, Hungary; 9Faculty of Materials and Chemical Sciences, University of Miskolc, H-3515 Miskolc, Hungary

**Keywords:** zinc, chemosensor, fluorescence, 2,2′-bipyridine, chelator, two-photon imaging, NMR, DFT

## Abstract

The few commercially available chemosensors and published probes for in vitro Zn^2+^ detection in two-photon microscopy are compromised by their flawed spectroscopic properties, causing issues in selectivity or challenging multistep syntheses. Herein, we present the development of an effective small molecular GFP chromophore-based fluorescent chemosensor with a 2,2′-bipyridine chelator moiety (GFZnP BIPY) for Zn^2+^ detection that has straightforward synthesis and uncompromised properties. Detailed experimental characterizations of the free and the zinc-bound compounds within the physiologically relevant pH range are presented. Excellent photophysical characteristics are reported, including a 53-fold fluorescence enhancement with excitation and emission maxima at 422 nm and 492 nm, respectively. A high two-photon cross section of 3.0 GM at 840 nm as well as excellent metal ion selectivity are reported. In vitro experiments on HEK 293 cell culture were carried out using two-photon microscopy to demonstrate the applicability of the novel sensor for zinc bioimaging.

## 1. Introduction

Metal ions are involved in many processes in living organisms, making their detection a principal task of modern analytical chemistry [1]. In particular, the detection of Zn^2+^ ions has attracted considerable research interest, as Zn^2+^ is the second most prevalent transition metal in the body, being endowed with significant roles in physiological processes [2]. Zn^2+^ is mainly present in a large variety of proteins, including enzymes. Moreover, biological processes such as cell division, metabolism, gene expression, and neurotransmission all require Zn^2+^ [3]. A reduced dietary intake of zinc correlates with several illnesses [4,5,6]. An imbalance of its homeostasis has also been associated with disorders such as Alzheimer’s and Parkinson’s disease, in addition to certain arthritic diseases [7,8,9,10]. The immunological relevance of Zn^2+^ has also been demonstrated recently, most notably in the connection between Zn^2+^ and SARS-CoV-2 infection [11].

The great importance of Zn^2+^ underlines the importance of having easily accessible analytical tools for Zn^2+^ detection, particularly in biological systems. Fluorescent methods provide a highly sensitive and accessible way to monitor Zn^2+^ both in space and time. Recently the use of fluorescence microscopy, and especially two-photon microscopy, has become widespread. The latter achieves excitation using high-intensity but lower-energy wavelengths, which enables the visualization of deeper tissues in 3D with localized excitation and less photodamage [12,13]. Fluorescence microscopy is the primary tool for studying metal ions in biological samples, and it requires fluorescent probes to detect the analyte. The availability of such probes can be a limiting factor in microscopy. Unfortunately, only a few fluorescent probes are available for two-photon imaging, and most of them are complex organic molecules with challenging multistep syntheses, making them hardly accessible for most users [14,15,16,17]. Typical Zn^2+^ sensors exhibit around a 10-fold fluorescence enhancement, the two-photon absorption values are in the 700–800 nm region, and most of them emit light at around 500 nm wavelength and have binding constants in the nanomolar concentration region. They have a few practical limitations, such as their solubility or selectivity [18]. In light of these aspects, as a continuation of our interest in novel fluorescent dyes and two-photon active compounds [19,20,21], we aimed to develop a novel family of two-photon sensors for Zn^2+^ by modifying the Green Fluorescent Protein (GFP) chromophore with a built-in 8-aminoquinoline motif, which made the probe Zn^2+^-sensitive [18]. The novel family, named GFZnP, exhibited excellent two-photon properties, its use was demonstrated in biological experiments, and it overcame the practical limitations of other sensors. However, due to the challenging synthesis of the 8-aminoquinalinde building blocks required for these sensors, they are still labor-intensive to access [18]. In this work, we aimed to prepare a similarly efficient sensor that is easily accessible in a single synthetic step.

The GFP chromophore exhibits intensive fluorescence inside the β-barrel of the GFP, where it is conformationally locked by hydrogen bonds (Figure 1A). However, in its bare form it loses the fluorescence due to non-emissive relaxation via intramolecular rotations [22]. Locking the conformation permanently or conditionally has been explored creatively to prepare various fluorescent dyes and sensor derivatives of the GFP chromophore [20,23,24,25].

The current work presents a usually applied zinc chelator, 2,2′-bipyridyl binding motif [26,27,28,29,30,31] attached to the GFP chromophore (Figure 1B), which is hypothesized to be nonfluorescent. However, we envision that Zn^2+^ binding to the 2,2′ bipyridine (BIPY) moiety would lock the conformation of the GFP chromophore and possibly disallow a photoinduced electron transfer (PeT) effect, resulting in a fluorescence turn-on. As (2,2′-bipyridine)-4-carbaldehyde is commercially available, the proposed probe would be accessible in a single Knoevenagel condensation, in line with the aim of this paper.

**Figure 1 ijms-25-03504-f001:**
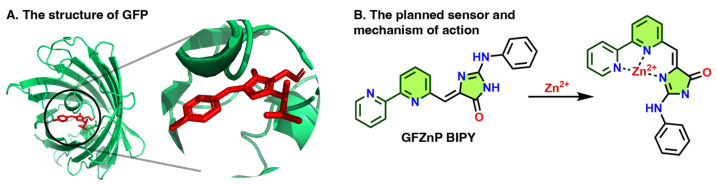
(**A**) Illustrative structure of the Green Fluorescent Protein (GFP), where its chromophore is highlighted in red [32]. (**B**) The planned sensor and its mechanism of action. The rings filled with green show the parts of the sensors resembling the GFP chromophore; the dark green lines represent the ionophore part.

## 2. Results and Discussion

The synthesis of GFZnP BIPY was carried out based on [2,2′-bipyridine]-4-carbaldehyde and the respective imidazolinones (Figure 1), which can be synthesized in a single step, as reported before [33]. After dissolving the two compounds in acetic acid, a catalytic amount of pyrrolidine was added, and the mixture was stirred at 110 °C for 1 h. After cooling to 5 °C, the product was precipitated from the reaction mixture, and the pure compound was isolated by simple filtration (>98%, HPLC). In the case of GFZnP BIPY, no further purification was needed, as opposed to the similar probes previously reported in the literature, which needed multiple synthetic steps and solvents and time-consuming purification with preparative HPLC.

The absorption and emission spectra were recorded in pH 7.4 HEPES buffer, which contained 2.75 µM of the prepared probe and either no Zn^2+^ (provided by the presence of 10 mM EGTA) or 1 mM Zn^2+^. The obtained spectroscopical properties of the prepared probe are summarized in Table 1. The prepared probe can be excited efficiently in the 400–450 nm range, which is compatible with the light sources of most one-photon microscope setups, and emits turquoise light (465–515 nm), with satisfactory brightness and an excellent fluorescence enhancement factor (FEF). The 52-fold FEF value is similar to the best measured FEF among our previously reported aminoquinoline-based sensor family. The turn-on of the sensor upon Zn^2+^ addition is instantaneous, and a full brightness is reached under a few seconds (Appendix A). Moreover, GFZnP BIPY has the same brightness and 1.5 times better FEF compared to GFZnP Pic, the brightest previously reported aminoquinoline, which had a four-step synthesis involving two chromatographies. Compared to these earlier probes, the absorption and emission wavelengths are ca. 20 nm blue shifted, which makes excitation possible using the blue channel light sources of microscopes instead of the green needed for previous sensors. Combined with the 70 nm Stokes shift, this feature enables the filtration of scattered excitation light without significant losses in the collection of fluorescence photons [34]. Any light source under 420 nm, including the very popular 405 nm laser, enables the complete detection of emitted light, as the emission spectrum does not overlap with this region. As shown in Figure 2A the absorption spectrum of GFZnP BIPY undergoes a 60 nm redshift during zinc binding, resulting in the final 422 nm absorption of the complex. This increases the contrast and fluorescence enhancement provided by the probe, as the free sensor does not undergo excitation at 422 nm since its highest wavelength absorption peak is centered around 362 nm in its free form. This spectral absorption shift is also visible to the naked eye, as the solution of GFZnP BIPY is colorless, but upon Zn^2+^ addition, it turns yellow. No such spectral redshift and no fluorescence signal was observed in a set of solvents with different polarity without Zn^2+^, confirming that the polarity of the environment does not interfere with the analyte sensing. However, a redshift and remarkable fluorescence was observed in glycerol, a highly viscous solvent. This is in agreement with our hypothesis that the conformational locking induces the fluorescence turn-on, since the conformational rotations are obstructed by the high viscosity, similarly to the ion binding (Appendix A). In summary, the probe presented herein offers comparable brightness and fluorescence enhancement to our best previously reported probes while also having a more optimal wavelength for most standard laser microscopes. It is accessible via a much simpler synthesis without the need for extensive purification methods, which makes it an excellent candidate for fluorescence microscopy and two-photon imaging [18].

After the promising initial results, the affinity of the probe towards Zn^2+^ was also determined via a competitive fluorescent titration experiment. The fluorescence intensities of different solutions containing EGTA and Zn^2+^ in varied ratios were plotted against the calculated free zinc ion concentration on a logarithmic scale (Figure 2B) to obtain a logistic plot. The midpoint of the fitted curve determined the apparent dissociation constant as *K*′_d_ = 6.61 nM. Fitting a complex model that assumes a 1:1 complex stoichiometry to the intensity values visualized against the free zinc concentration (Figure 2C) provided a similar value, *K*′_d_ = 6.23 nM. Both suggest that GFZnP BIPY forms a highly stable complex with zinc, similar to the best sensors reported, making it excellent for imaging [18,35]. The 1:1 stoichiometry of this complex was also confirmed by a fluorescence Job’s plot (Figure 2D). Similarly sized chelators with 2,2′-bipyridyl motifs also form 1:1 complexes, according to previous studies [26,27,30,31,36].

A molecular-level investigation through ^1^H 
NMR spectroscopy also provided clear evidence of complexation between GFZnP 
BIPY and Zn^2+^ ions. At room temperature, in DMSO-*d*_6_, 
the probe exhibits broad signals, indicating a dynamically changing molecular structure 
with conformational rates in the intermediate range (i.e., time constant on the 
ms scale) relative to the NMR timescale (Appendix A). 
The addition of Zn^2+^ ions (in the form of Zn(OTf)_2_) 
caused substantial chemical shift changes and had a significant narrowing 
effect on the formerly broad signals. This indicates that complex formation 
blocks the free rotation of the ligand, giving rise to a rigid, non-fluxional 
coordinative compound, as we hypothesized. At high Zn^2+^ 
concentrations (>1 equivalent, Appendix A), 
the well-resolved narrow peaks of the formed complex enabled comprehensive 
characterization through 2D measurements (see Appendix A for details). In the initial titration region, we observe the 
formation of a new peak series (
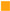
), which are narrower compared 
to those of the free probe (
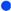
)) but different from the peaks 
of the final, stable complex (
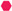
)). We propose that at low Zn^2+^ 
concentrations, one Zn^2+^ ion can bind to more than one ligand, 
resulting in a complex with a stoichiometry of M:L = 1:2, for example. This 
complex has a lower stability than the final, 1:1 complex since, at 100% 
titration, the ratio of the intermediate and final complex is 1:4, and the 
final complex can only be observed by two equivalents (Figure 3A,B). Theoretical computations (see Appendix A) also confirmed similar complex structure formation. Without Zn^2+^, 
the free probe can undergo conformational rotations (Appendix A). Zn^2+^ binding by the 
sensor and the deprotonation of the formed complex—also observed in NMR 
spectroscopy—are found to be energetically favorable.

Finally, after the detailed characterization presented above, the selectivity of the sensor was tested to confirm that the strong binding is only applicable to zinc. The GFZnP BIPY probe did not emit considerable fluorescence in the presence of other biologically relevant metal ions (Figure 4A). The strongest interference was in the cases of Ca^2+^ and Mg^2+^ under a 20% fluorescent signal for both compared to Zn^2+^ only when present in a very high, 1 mM concentration. Therefore, the possibility of an interference in a real-world application is unlikely, as a 1 mM Ca^2+^ concentration would give a similar signal to a 1 nM Zn^2+^ concentration. Cd^2+^ also had a similarly slight interference but one that is even less concerning for biological applications. Electron scavenger ions such as Fe^3+^ and Cu^2+^ quenched the fluorescence of the probe when present in equimolar quantities, which may lead to false-negative signals if they are present. However, in biological use cases, this is rarely a problem. No interference was observed in the anion screening experiments (Figure 4B). In general, the selectivity of the probe was excellent, further illustrated by the photograph shown in Figure 4A. The pH sensitivity of GFZnP BIPY was tested between pH 3 and 9. The probe does not give false positives for pH changes. However, very acidic pH levels (>pH 5) quench the fluorescence, which needs to be taken into account in the case of specific uses (e.g., acidic vesicles, lysosomes). Although the fluorescence is quenched by the protonation of the complex, the Zn^2+^ binding is still present even at low pH levels, as the UV-spectra of acidic solutions containing Zn^2+^ differ from zinc-free spectra at different pH levels (Appendix A). The fluorescence was relatively stable in a wide window around the physiological pH level (Figure 4C). The water solubility of the probe was found to be 73 µM (25 mg/L, Appendix A). The bench stability of the probe was studied using a 5 mM solution in DMSO/EtOH 1:1. No decomposition was observed after 4 months of storage at 5 °C. (Appendix A). The use of GFZnP BIPY for determining quantitative Zn^2+^ detection was demonstrated by recording a calibration line of total Zn^2+^ content using higher probe concentrations (Appendix A). The calibration was linear in the 0–20 µM range using a 33 µM probe concentration. The method exhibited good performance with a limit of detection of 129 µg/L. 

With the spectroscopic and Zn^2+^ binding properties in hand, the applicability of GFZnP BIPY in the proposed main application, two-photon fluorescence microscopy, was demonstrated. First, the two-photon action cross section of the probe was determined in the presence and absence of Zn^2+^. Almost no two-photon fluorescence could be detected in the solution that did not contain Zn^2+^, whereas a strong signal was emitted by the one that did (Figure 5A). The two-photon fluorescence enhancement (*FEF*_2P_) was 100-fold, while the peak action cross section was *δ Φ* = 3 GM, both similar to those of previously reported state-of-the-art probes. The two-photon spectrum coincides with the wavelength-doubled single-photon absorption spectrum with large precision (in the presence of Zn^2+^), which suggests that the same electronic transition is predominant during two-photon excitation (Figure 5B). This results in a two-photon excitation peak at 850 nm, which is ideal for the Ti/sapphire pulsed lasers most commonly used in two-photon setups. To demonstrate this, HEK 293 cells were incubated with GFZnP BIPY for 5 min in a 10 µM concentration, and two-photon images were recorded (Figure 5C). No colocalization (*p* < 0.1) with the nuclei, the membrane, or mitochondria were observed (Appendix A). Then, Zn^2+^ were added and imaging continued. An immediate strong increase in the fluorescence was observed (Figure 5D) at the parts of the image covered with cells. In randomly selected fields of view containing cells, the increase in fluorescence was 6.7-fold (Figure 5F), which was stable over the course of the 10 min imaging period. No cytotoxicity was observed in a standard cytotoxicity assay on HEK 293 cells (Appendix A). To prove that the sensing is reversible, a strong Zn^2+^ chelator (TPEN) was added to the cells to displace Zn^2+^ from its complex with GFZnP BIPY. Indeed, TPEN addition resulted in the quenching of the fluorescence (Figure 5E). This experiment is strong proof that GFZnP BIPY can be efficiently applied for the real-time monitoring of Zn^2+^ in biological samples.

## 3. Materials and Methods

### 3.1. Materials

Reagents and solvents were purchased from Merck (Budapest, Hungary) and used as received. For spectroscopy, UVASol solvents (Supelco, Merck, Budapest, Hungary) were used. Deionized water was prepared in house using a Milli-Q RiOs-DI-3UV purifier, and resistivity was always kept >10 MΩ cm. Compound purity was confirmed by HPLC-MS using a Shimadzu LC-40D XR system equipped with a SIL-40C XR autosampler, SPD-M40 photodiode array detector, and an RF-20A XS fluorescent detector coupled to an LCMS-2020 DUIS Mass Spectrometer operated in alternating negative and positive modes. Separation was performed on an Ascentis Express C18, 2 μm UHPLC column (L × I.D. 5 cm × 2.1 mm) operated at 40 °C using a CTO-40s column oven. Analytical separation was achieved by a linear gradient elution in 5 min using 0.1% *v*/*v* TFA in water (A) and 0.1% *v*/*v* TFA in MeCN (B). NMR spectra were recorded on a Varian Unity INOVA spectrometer operating at an equivalent ^1^H frequency of 400 MHz. Spectra were acquired at room temperature unless noted otherwise. Notations for the ^1^H NMR spectral splitting patterns include singlet (s), doublet (d), triplet (t), broad (br), and multiplet/overlapping peaks (m). Chemical shifts are given as *δ* values in ppm, and coupling constants (*J*) are expressed in Hertz. Accurate mass measurements were carried out on a high-resolution Thermo Fisher Scientific Q-Exactive Focus hybrid quadrupole–orbitrap mass spectrometer used with a heated electrospray ionization source. Samples were dissolved in an acetonitrile/water 1:1 (*v*/*v*) solvent mixture containing 0.1% (*v*/*v*) formic acid. Flow injection analysis was performed using a 50 µL min^−1^ eluent flow provided by a Thermo Scientific UPLC. Spectroscopic studies were realized using a Shimadzu UV-1900i spectrophotometer and an RF-6000 spectrofluorometer using 1.0 cm path length quartz cell. Absorbances were measured with a 1 nm slit width in the high-speed mode of the instrument. The fluorescence spectra were recorded at 2000 nm min^−1^ scan speed, using 10 nm emission and excitation slit widths by excitation at the absorption maximum of the Zn^2+^ complex. Background correction of the solvent was applied to the UV-VIS spectra. Two-photon experiments were carried out with a Femtonics FemtoSMART-Dual microscope using a Nikon CFI75LWD 16× objective (*NA* = 1.0) and a tunable high-power Coherent Chameleon Discovery Ultra II Ti:Sapphire laser (*λ*_ex_ = 700–1040 nm). The emitted light was detected using a Hamamatsu H11706-40 photomultiplier tube, recorded by Femtonics (Budapest, Hungary) MES 6.5.8966 software running on MATLAB 2017a, and analyzed in ImageJ (https://imagej.net/ij/).

### 3.2. Synthesis of GFZnP BIPY

A total of 100 mg (0.54 mmol) 2,2′-bipyridine-4-carbaldehyde was dissolved in 3 mL acetic acid, and 95 mg (0.54 mmol) 2-(phenylamino)-3,5-dihydro-4*H*-imidazol-4-one (prepared as previously reported [33]) was added to the solution. One drop of pyrrolidine was added, and the mixture was stirred at 110 °C for 1 h. The reaction mixture turned into a greenish brown color, and HPLC-MS analysis indicated a quantitative conversion. The mixture was cooled to room temperature, and the product was precipitated. After filtration and washing with ethanol and copious amounts of deionized water, an off-white powder was isolated. Yield: 70 mg, 40%. Due to the internal dynamics of the product, its characterization in NMR spectroscopy was challenging, as broad peaks were observed. Therefore, we characterized the NMR spectrum of the Zn^2+^-complex of GFZnP BIPY, which, as presented, has sharp peaks that are in accordance with the expected product (Appendix A).

GFZnP BIPY: ^1^H NMR (400 MHz, DMSO-*d*_6_) *δ* 9.71 (brs, 1H), 8.68 (brd, 1H), 8.46 (brs, 1H), 8.23 (brs, 1H), 7.99 (brs, 2H), 7.52 (brd, 5H), 7.14 (brs, 1H), 6.52 (brd, 1H).

HRMS (ESI+) *m*/*z* calcd. for C_20_H_15_N_5_O^+^ [M+H]^+^: 342.1350. Found: 342.1342 *δ* = −2.28.

GFZnP BIPY Zn^2+^-complex: ^1^H NMR (400 MHz, DMSO-*d*_6_) *δ* 10.03 (s, 1H), 8.83–8.76 (m, 2H), 8.64 (d, *J* = 7.8 Hz, 1H), 8.40 (td, *J* = 7.8, 1.7 Hz, 1H), 8.36 (t, *J* = 8.0 Hz, 1H), 8.10 (d, *J* = 7.8 Hz, 1H), 7.96–7.88 (m, 1H), 7.55 (dd, *J* = 8.4, 7.2 Hz, 2H), 7.48–7.44 (m, 2H), 7.45–7.39 (m, 1H), 6.83 (s, 1H). ^13^C NMR (101 MHz, DMSO-*d*_6_) *δ* 169.23, 160.96, 153.56, 148.95, 148.76, 148.35, 142.12, 142.02, 140.24, 134.56, 130.61, 128.10, 127.82, 124.84, 123.84, 121.87, 107.66.

^15^N NMR (41 MHz, DMSO-*d*_6_) *δ* −127.74, −124.79, −229.28, −273.51.

### 3.3. Spectroscopical Characterization

The measurements were carried out in aqueous buffers by diluting a concentrated dye stock solution prepared as follows: 2.06 mg of GFZnP BIPY was dissolved in 10 mL DMSO to obtain a 600 μM stock solution. A total of 3 mL of the corresponding buffer solution was spiked with 20 μL of this stock solution to achieve a final dye concentration of 4 μM in each measurement and a DMSO concentration of 0.66%. To measure the fluorescence of the free probe and the Zn^2+^ complex buffer solutions were prepared by dissolving 720 mg EGTA (10 mM) or 72.7 mg Zn(OTf)_2_ (1 mM) in 200 mL deionized water containing 2.38 g HEPES and adjusting the pH to 7.4 by adding concentrated NaOH solution. The molar absorption coefficient was determined using the Lamber–Beer law, as shown in Equation (1):*ε*/(dm^3^(mol cm)^−1^) = *A c*^−1^*l*^−1^(1)

The fluorescence enhancement factor was calculated based on Equation (2), where *F* and *F*_0_ are the areas under the emission curve of the GFZnP BIPY–Zn^2+^ complex and free probe, respectively. The fluorescence quantum yield was determined using the standard relative method reported in the literature. Fluoresce in 0.1 M NaOH was used as a standard (*Φ*_ref_ = 0.95), and Equation (3) was used to determine the quantum yield of GFZnP BIPY (*Φ*). The subscript *s* denotes the sample, *ref* denotes the reference values, *F* stands for the areas under the fluorescence emission spectra, *A* stands for the absorbance of the measured samples, and *n* marks the refractive index of the solvents used [37].
*FEF* = (*F* − *F*_0_)/*F*_0_
(2)
(3)$$\phi =\phi_{ref}\times \frac{\int F_{s}}{\int F_{ref}}\times \frac{A_{ref}}{A_{s}}\times \frac{n_{s}^{D}}{n_{ref}^{D}}$$

Direct titration of the presented probe showed only a saturation of the used 4 μM sensor concentration; therefore, a lower *K′*_d_ was suspected, and a competitive titration was employed to determine the value. A 1 mM EGTA solution was prepared (72 mg EGTA and 2.38 g HEPES in 200 mL deionized water, pH adjustment to 7.4 by conc. NaOH). To 100 mL of this EGTA solution, 36.35 mg Zn(OTf)_2_ was added, yielding a solution containing 1 mM Zn^2+^ and 1 mM EGTA. Next, 16.5 μL dye stock solution was added to 2.5 mL of the 1 mM EGTA solution, and 133 μL dye stock was added to 20 mL of the 1 mM Zn^2+^-EGTA solution, resulting in a 6.6 μM dye concentration in both cases. Then, the spectrum of the only EGTA-containing solution was recorded, and small aliquots of the solution were exchanged with the Zn^2+^-EGTA solution while recording the spectrum in each step. As EGTA and Zn^2+^ are both in large excess concentrations compared to our dye, the free zinc concentration in the solution was determined based on the stability of the Zn^2+^-EGTA complex. Given, that the apparent stability of the Zn^2+^-EGTA complex is log(*K*′_ZnEGTA_) = −8.97, Equation (4) can be used to calculate the free zinc concentration.
(4)$${\left[{Zn}^{2+}\right]}_{free}=\frac{{\left[{Zn}^{2+}\right]}_{total}}{{\left[EGTA\right]}_{free}\;{K\prime }_{ZnEGTA}}$$

The integrated intensities were plotted against log[Zn^2+^]_free_, and the midpoint of the logistic curve was taken as the *K*′_d_ of GFZnP BIPY. Alternatively, a 1:1 complex model (Equation (5), where *F* is the integrated fluorescence intensity and *F*_max_ and *F*_0_ are the integrated fluorescence intensities of the complex and free probe, respectively) was also fitted to the plot of the integrated fluorescence intensity against [Zn^2+^]_free_ to confirm the initial value.
(5)$$F=\left[1+\left(\frac{F_{max}}{F_{o}}-1\right)\times \left(\frac{{\left[{Zn}^{2+}\right]}_{free}}{{{K\prime }_{ZnEGTA}+[{Zn}^{2+}]}_{free}}\right)\right]\times F_{o}$$

The binding ratio was determined using a fluorescence Job’s plot. Solutions containing the studied probe and Zn^2+^ in different molar ratios were prepared by the addition of small aliquots (<30 μL) of the dye stock solution and the 1 mM Zn^2+^-containing buffer to 3 mL of 50 mM HEPES pH 7.4 buffer. In each sample, the total concentration of GFZnP BIPY and Zn^2+^ was kept at *c*_Zn_ + *c*_probe_ = 10 μM.

The NMR titration experiment was carried out by sequentially adding a 33 mM solution of Zn(OTf)_2_ in DMSO-*d_6_* to a 5 mM solution of GFZnP BIPY in DMSO-*d_6_.*

The pH sensitivity was characterized by measuring the fluorescence intensity and UV-VIS of GFZnP BIPY in 50 mM acetate (pH = 3.5–5.0) and HEPES (pH 6.0–9.0) buffers containing either 5 mM EGTA or 1 mM Zn(OTf)_2_. Selectivity was determined by recording the fluorescence intensity in 10 mM solutions of the following salts in pH 7.4 HEPES (50 mM): cations: CaCl_2_, MgCl_2_, NaCl, KCl, Fe(NO_3_)_2_, CuSO_4_, SnCl_2_, Cd(NO_3_)_3_, Pb(NO_3_)_2_, and Zn(OTf)_2_; anions: NaCl, KBr, KI, KF, NaOAc, K_2_SO_4_, Na_2_HPO_3_, KNO_3_, 4-aminobutyric acid, and L-glutamic acid. Intensity values were normalized to the Zn(OTf)_2_-containing solution, and the free sample contained only pH buffer. Interference measurements were repeated in similar solutions also containing 10 mM Zn(OTf)_2_.

The turn-on rate of the sensor was measured by recording the fluorescence of a 3 mL HEPES pH 7.4 solution containing 4 μM GFZnP BIPY and suddenly adding 30 μL of 100 mM Zn(OTf)_2_ solution to reach a final Zn^2+^ concentration of 1 mM. The excitation and emission wavelength was set to the respective maxima, and the slit widths were 3.0 nm. The instrument was set to a 500 ms accumulation time, and sampling was carried out at a 1 Hz rate.

The water solubility of GFZnP BIPY was determined by preparing a 100 mM solution in DMSO and adding 1 µL of the solution to 3 mL HEPES pH 7.4 buffer (0.03% DMSO content). The precipitated mixture was filtered through a 0.45 µM PES syringe filter to obtain a saturated aqueous solution. The absorbance of this solution was measured to determine the concentration of the saturated solution based on Equation (1). 

To use GFZnP BIPY for the quantitative determination of Zn^2+^ in aqueous samples, a calibration line was recorded by adding known total aliquots of Zn^2+^ from a 2 mM aqueous stock solution to HEPES pH 7.4 buffered solution of the probe (33.3 µM). In total, 50 µL of Zn^2+^ stock was added in aliquots of 3 mL; dilution was not taken into account. A calibration line was fitted. The performance of the method was described by the LOD, which was calculated based on Equation (6), where *σ* is the standard deviation of the blank measurement, and *s* is the slope of the calibration curve:*LOD*/µM = (3*σ* + *F*_blank_)/*s*
(6)

The two-photon cross section (*δ,* in Goepper-Mayer units, noted as GM) was measured using a relative method similar to the one used for quantum yield determination [38]. Rhodamine 6G in MeOH (20 μM) was used as a standard. A 120 μM solution of GFZnP BIPY was prepared in zinc-free and 0.8 mM Zn^2+^-containing pH 7.4 HEPES buffers (50 mM, containing 10% DMSO). The samples were loaded into capillaries and placed into the field of view of the two-photon microscope. The incident light was focused into the capillary and the average intensity was recorded (*F*^2P^) while the laser light was kept at a constant 15 mW power and the excitation wavelength was increased in 10 nm steps between *λ*_ex_ = 750 and 1040 nm. The emitted light was detected between 475 nm and 575 nm, which was corrected based on the one-photon spectrum as shown in Equation (7). Here, ref and s subscripts stand for sample and reference; *c*, *n*, and *Φ* mark the concentration, refractive index, and the one-photon quantum yield, respectively. The integral ratios are the relative amount of emitted light cut off by the filter based on the 1P emission spectra.
(7)$$\delta =\delta_{ref}\times \frac{F_{s}^{2P}\cdot c_{ref}\cdot n_{ref}^{D}}{F_{ref}^{2P}\cdot c_{s}\cdot n_{s}^{D}}\times \frac{{\left(\int_{475}^{575}F/\int F\right)}_{ref}}{{\left(\int_{475}^{575}F/\int F\right)}_{s}}\times \phi_{S}^{-1}$$

### 3.4. Biological Validation

HEK 293 cells (CRL-1573, ATCC) cultured in Dulbecco’s Modified Eagle’s Medium (DMEM) supplemented with 10% fetal bovine serum (FBS), 100 U/mL penicillin, and 100 μg/mL streptomycin were incubated (37 °C with 5% CO_2_) in a humidified incubator and subcultured every 2–3 days. The cultures were plated onto polylysinated glass coverslips at a density of approximately 26,300 cells/cm^2^ before further incubation for two more days. The cells were stained with a 10 μM GFZnP BIPY solution containing the following: 140 mM Na-gluconate, 5 mM K-gluconate, 3 mM CaCl_2_, 1 mM MgCl_2_, 5 mM D-glucose, and 10 mM HEPES adjusted to pH 7.4 using NaOH. The probe was added from a 5 mM DMSO-based stock solution, and staining for 5 min was carried out immediately before the measurements. After staining, the solution was changed to a fresh one and the coverslip was placed into the recording chamber. Baseline images were recorded; then, 10 μM Zn^2+^ pyrithione was added to the immersion from a concentrated stock solution and recording continued. Then, 100 mM *N*,*N*,*N*′,*N*′-tetrakis(2-pyridylmethyl)ethylenediamine (TPEN) was added to the immersion and imaging continued. Images were recorded every ca. 30 s. 

The cytotoxicity of GFZnP BIPY was studied with Hoechst 33342 and Propidium iodide stainings. The HEK293 cells were investigated in the wells of a 24-well plate at 10× magnification using an EVOS M5000 Imaging System (ThermoFisher). Four wells were dyed with GFZnP BIPY for 5 min in the same solution, as in the two-photon experiments, and four wells were left as controls. The solution was then changed, and the plates were incubated for 30 or 60 min. Images were taken again, and every well was stained with Hoechst 33342 (10 µg/mL) and Propidium iodide (1 µg/mL) in phosphate-buffered saline (PBS). The plates were incubated for 20 min, then imaged again. Propidium iodide staining was detected in the red channel (585/628 nm), and Hoechst staining was detected in the blue channel (357/447 nm). The composited images are shown in Appendix A. The ratio of propidium iodide-stained and Hoechst-stained cells gave the viability value, as propidium iodide stains only dead cells, whereas Hoechst stains all cells. Cell counting was performed with ImageJ. For each image, a threshold was set (55–255 for blue channel; 61–255 for red channel). For the blue channel images, watershed processing was applied to count conjoined cells more accurately. Then, objects with 100 px^2^ area and a roundness above 0.3 were automatically counted. The cell counts and viabilities are summarized in Appendix A. A colocalization study was carried out similarly using the same microscope setup and staining procedure with GFZnP BIPY and Hoechst. However, instead of propidium iodide, Mitotracker CMXRos (ThermoFisher) was used in a 100 nM concentration for labeling the mitochondria (detected in the red channel), and 100 nM MemBright-Cy5 was used for labeling the cell membranes detected in the magenta channel (635/692 nm). 

### 3.5. Computational Methods

Theoretical calculations were carried out with Gaussian16, Revision C.01 software [39] using the standard convergence criteria given as default. Optimization and vibrational frequencies were calculated using the M06-2X method [40] using the 6-311++G(2d,2p) basis set and the IEFPCM method (ε = 78.3553 for water) [41,42]. Thermodynamic functions were computed at 298.15 K. For wavelength prediction, vertical excitation was modeled by the TD-B3LYP/6-311++G(2d,2p)//PCM(water) level of theory using the geometries optimized at B3LYP/6-311++G(2d,2p)//PCM(water). The emission wavelengths were calculated after optimization using the geometries provided by TD-B3LYP/6-311++G(2d,2p)//PCM(water).

## 4. Conclusions

In this work, we presented a novel two-photon fluorescent Zn^2+^ sensor for application in biology based on a modified GFP chromophore containing a 2,2′-bipyridyl binding motif. The novel probe presented herein showcases a 53-fold fluorescence increase, a 421 nm absorption wavelength with 492 nm emission, and a 3 GM two-photon cross section triggered by nanomolar concentrations of zinc ions. The Zn^2+^ binding results in a rigid 1:1 chelate complex with a locked conformation, which induces the fluorescence turn-on. The probe performs similarly to the state-of-the-art sensors; however, its synthesis is straightforward, making it accessible for zinc imaging on biological samples.

## Data Availability

The authors declare that all the data supporting the findings of this study are available within the paper and the Appendix A. Additional raw data are available from the corresponding authors upon request.

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
