# Peer review of "A Molecular Hybrid of the GFP Chromophore and 2,2′-Bipyridine: An Accessible Sensor for Zn2+ Detection with Fluorescence Microscopy"

_ijms, 2024, doi:10.3390/ijms25063504_

Round 1
Reviewer 1 Report
Comments and Suggestions for Authors
The article "A molecular hybrid of the GFP chromophore and 2,2′-bipyridine: an accessible sensor for Zn2+ detection with fluorescence microscopy" introduces a highly sensitive fluorescent method for detecting zinc(II) using GFZnP BIPY. The method relies on turn on of the chemosensor's fluorescence by Zn2+ in solution. While the research is of high quality, there are areas that need improvement, as detailed below:
1. Please include error bars in Figure 4A.
2. What causes the decrease in fluorescence intensity of the zinc complex at acidic pH? (Figure 4b)
3. What is the detection limit for zinc ions using the chemosensor?
4. Anions of different salts such as chlorides, nitrates, OTf- were used in the work, what is the effect of the anions on the detection of zinc ions?
5. Why was the M06-2X functional chosen for quantum chemical calculations?
6. In lines 204 – 205, it is mentioned that "No decomposition was observed after 6 months of storage at 5°C. (Figure S12)". However, in Figure S12's caption, it states "HPLC-DAD chromatogram of GFZnP BIPY detected in the 350 - 450 nm absorption range, after 4 months of storage at room temperature in a 5 mM DMSO:EtOH 1:1 solution." When were these measurements taken?
7. What was the reason for using different basis sets to calculate geometries and vertical excitation?
Author Response
The article “A molecular hybrid of the GFP chromophore and 2,2′-bipyridine: an accessible sensor for Zn2+ detection with fluorescence microscopy” introduces a highly sensitive fluorescent method for detecting zinc(II) using GFZnP BIPY. The method relies on turn on of the chemosensor’s fluorescence by Zn2+ in solution.
We thank the reviewer for the thorough review and overall positive evaluation. Each point has been addressed, and further experiments have been carried out to improve the manuscript in line with the suggestions. Please find our replies below.
While the research is of high quality, there are areas that need improvement, as detailed below:
- Please include error bars in Figure 4A.
The cation selectivity was originally measured in a single instance. The measurements have now been repeated and the data is recalculated based on triplicate measurements. Error bars representing the standard deviation of the new measurements have been added to Figure 4A.
- What causes the decrease in fluorescence intensity of the zinc complex at acidic pH? (Figure 4b)
The decrease in fluorescence intensity at acidic pH could in theory be either caused by the protonation of binding nitrogen atoms and thus the loss of Zn2+ binding, or the protonation of the complex without dissociation, which results in a non-fluorescent complex. In a new experiment, the UV-VIS spectra of Zn2+-free and Zn2+ containing solutions of GFZnP BIPY at different pH were recorded. The results showed that the UV-spectra of Zn2+-free solutions highly differ from the Zn2+ containing solutions at all pH values. In light of this we infer that the zinc complex does not dissociate upon protonation at acidic pH, only the fluorescence is quenched due to electronic effects. The results are now shown in Fig. S15 of the supplementary information and are now discussed in the manuscript.
- What is the detection limit for zinc ions using the chemosensor?
The limit of detection is not an intrinsic property of a chemosensor, rather it depends on the analytical assay, the instrumentation and even the applied concentration of the chemosensor. In bioimaging, the concentration of the sensor, the analyte and the matrix effects significantly differ between regions even of a single image. Therefore, quantification and detection limit calculation are not possible for an intensity change fluorescence probe, such as GFZnP BIPY, in bioimaging application. However, a simple analytical method for detecting zinc in aqueous samples with GFZnP BIPY and fluorometer detection has been developed and the limit of detection has been established, which is 129 µg/L or 1.98 µM. Further analytical development may be able to improve these parameters for such applications.
- Anions of different salts such as chlorides, nitrates, OTf- were used in the work, what is the effect of the anions on the detection of zinc ions?
A comprehensive screening of biologically relevant anions has been included in the manuscript. No turn-on was detected by the analysed anions and none of them interfered with the Zn2+ ion detection. Figure 4 has been extended with a new panel showing the results of these measurements.
- Why was the M06-2X functional chosen for quantum chemical calculations?
The M06-2X functional was used for geometry optimization as it gives very accurate results for non-covalent interactions (Zhao and Truhlar, Theor. Chem. Account. 2008, 120, 215.), such as hydrogen bonding or zinc chelation in our case. However, B3LYP has been found more accurate for the calculation of vertical excitation energies (Jacquemin et al. J. Chem. Theory Comput. 2010, 6, 2071.) Therefore, we used B3LYP to calculate excitation energies and wavelengths for M06-2X optimized geometries to get accurate computational data. This methodology was also successfully applied in our previous work on a number of structurally similar Zn2+ sensors to get excitation and emission wavelength in good agreement with the experimental results (Csomos et al. Sens. Actuators B Chem. 2024, 398, 134753.).
- In lines 204 – 205, it is mentioned that "No decomposition was observed after 6 months of storage at 5°C. (Figure S12)". However, in Figure S12's caption, it states "HPLC-DAD chromatogram of GFZnP BIPY detected in the 350 - 450 nm absorption range, after 4 months of storage at room temperature in a 5 mM DMSO:EtOH 1:1 solution." When were these measurements taken?
We would like to thank the reviewer for raising our attention to this. The measurements were taken after 4 months. This error has now been corrected in the manuscript.
- What was the reason for using different basis sets to calculate geometries and vertical excitation?
We thank the Reviewer for noticing this discrepancy. Initial calculations were carried out with a simpler 6-31G(d,p) basis set, which were then refined with a larger 6-311++G(2d,2p) basis set. However, the submitted manuscript erroneously contained the primary data computed at the lower basis. All the presented computational data has been updated to reflect the results obtained with the larger basis set 6-311++G(2d,2p). The results and discussion presented in the manuscript is reflective of the data obtained with 6-311++G(2d,2p).
Reviewer 2 Report
Comments and Suggestions for Authors
In this manuscript, a new two-photon fluorescent Zn2+ sensor for biological applications has been developed. This sensor is based on a modified GFP chromophore containing a 2,2'-bipyridyl binding motif and exhibits a 50-fold increase in fluorescence in response to nanomolar concentrations of zinc ions. The authors propose the formation of a rigid 1:1 chelate complex that turns on fluorescence upon binding with Zn2+. This probe potentially allows the detection of Zn2+ with high sensitivity and selectivity, and promises to be a useful tool for biological research and medical applications. The experiments were successfully performed and there are no problems with the data obtained. However, there are several important aspects where data or details are missing in the manuscript, which are highlighted below.
1. Binding and dissociation rate constants:
The manuscript lacks direct data on the binding rate constant (kon) and dissociation rate constant (koff) between GFZnP-BIPY and Zn2+ ions. These constants are usually determined by experimental kinetic studies and are critical for assessing the reactivity, sensitivity, and specificity of the sensor for specific ions. We recommend adding experimental measurements and explanations of applicable models.
2. Cytotoxicity:
No experimental cytotoxicity results are provided. For the sensor to be used in practical biological applications, its safety must be established. We recommend providing cytotoxicity test data using standard cell lines such as HEK 293 cells.
3. Water solubility:
Data on the water solubility of the sensor is missing. This information is critical to the scope and method of use of the sensor, so the addition of solubility test results is desirable. Water solubility is particularly important when considering biological applications, as insufficient solubility may increase cytotoxicity or result in uneven distribution within cells.
4. Lipophilicity and cell membrane accumulation:
The manuscript does not discuss the potential impact of the high lipophilicity of the sensor on cell membrane accumulation and its ability to bind to zinc ions. Cell membrane accumulation may limit access to zinc ion binding sites and prevent the probe from adequately reacting with free zinc ions in the cell. In order to accurately understand the distribution and function of the sensor within the cell, experiments are needed to assess its subcellular localization and binding efficiency with zinc ions. This could be achieved by using high resolution fluorescence microscopy imaging to observe the distribution of the probes within the cell. The reviewer recommends providing experimental data on intracellular localization and discussing methods to address this issue.
5. Fluorescence in various organic solvents:
The manuscript does not mention the need to measure fluorescence properties in different organic solvents. Evaluating the fluorescence characteristics of the sensor in different polar and non-polar organic solvents is useful to assess how the sensor performs in real samples and biological environments. In addition, solvent-dependent stability information can help select suitable solvents for long-term storage or use. Data on fluorescence properties in different solvents can deepen our understanding of the application range and behavior of the sensor.
Author Response
In this manuscript, a new two-photon fluorescent Zn2+ sensor for biological applications has been developed. This sensor is based on a modified GFP chromophore containing a 2,2'-bipyridyl binding motif and exhibits a 50-fold increase in fluorescence in response to nanomolar concentrations of zinc ions. The authors propose the formation of a rigid 1:1 chelate complex that turns on fluorescence upon binding with Zn2+. This probe potentially allows the detection of Zn2+ with high sensitivity and selectivity, and promises to be a useful tool for biological research and medical applications. The experiments were successfully performed and there are no problems with the data obtained. However, there are several important aspects where data or details are missing in the manuscript, which are highlighted below.
We thank the reviewer for the thorough review and overall positive evaluation. Each point has been addressed, and further experiments have been carried out to improve the manuscript in line with the suggestions. Please find our replies below.
- Binding and dissociation rate constants:
The manuscript lacks direct data on the binding rate constant (kon) and dissociation rate constant (koff) between GFZnP-BIPY and Zn2+ ions. These constants are usually determined by experimental kinetic studies and are critical for assessing the reactivity, sensitivity, and specificity of the sensor for specific ions. We recommend adding experimental measurements and explanations of applicable models.
We appreciate the recommendation of the reviewer. Initially, we did not measure the binding rate of the sensor, as the binding seemed instantaneous upon visual inspection during our measurements. We have now carried out a time course measurement in a fluorometer, where Zn2+ is added to the solution while the fluorescence signal is continuously recorded. The complete turn-on is achieved in under 5 seconds (only 4 transient data points), which is comparable to the mixing speed of the concentrated zinc solution with our sample. Therefore, an exact binding rate constant cannot be reliably calculated from this data. Nonetheless, the time scale of fluorescence turn-on appears to be shorter than a second, which makes the probe suitable for biological applications. A figure showing the time course measurement has been added to the supplementary information and the manuscript now discusses the turn-on speed.
- Cytotoxicity:
No experimental cytotoxicity results are provided. For the sensor to be used in practical biological applications, its safety must be established. We recommend providing cytotoxicity test data using standard cell lines such as HEK 293 cells.
No cytotoxicity was observed during the in vitro tests reported in our work. However, as recommended by the Reviewer, we have carried out cytotoxicity tests on a standard HEK293 cell line. Cells were stained with GFZnP-BIPY and after 30 or 60 minutes, Hoechst 33342 and propidium iodide nucleus staining was also applied. The total cell count was established based on the Hoechst staining and the dead cell count was based on the propidium iodide staining. No change in the cell viability was observed in comparison with control wells that were not stained with GFZnP-BIPY, suggesting that our dye is not cytotoxic under the applied experimental conditions. This experimental data has been added to the supplementary information and the manuscript.
- Water solubility:
Data on the water solubility of the sensor is missing. This information is critical to the scope and method of use of the sensor, so the addition of solubility test results is desirable. Water solubility is particularly important when considering biological applications, as insufficient solubility may increase cytotoxicity or result in uneven distribution within cells.
We agree with the Reviewer, that water solubility of the probe is an important aspect. We have determined the water-solubility of GFZnP BIPY and found it to be 73 µM, satisfactory for the desired applications. We have added the new data to the manuscript and the supplementary information. We did not observe cytotoxicity or membrane localization upon application.
- Lipophilicity and cell membrane accumulation:
The manuscript does not discuss the potential impact of the high lipophilicity of the sensor on cell membrane accumulation and its ability to bind to zinc ions. Cell membrane accumulation may limit access to zinc ion binding sites and prevent the probe from adequately reacting with free zinc ions in the cell. In order to accurately understand the distribution and function of the sensor within the cell, experiments are needed to assess its subcellular localization and binding efficiency with zinc ions. This could be achieved by using high resolution fluorescence microscopy imaging to observe the distribution of the probes within the cell. The reviewer recommends providing experimental data on intracellular localization and discussing methods to address this issue.
We thank the Reviewer for this remark. We did not observe problems with the reaction of the probe with zinc, and our images showed rather a dotted pattern, which may indicate the natural localization of zinc inside the cells in addition to the localization of the probes. A vesicular/lysosomal localization is probable. According to the recommendation we have carried out a colocalization study and did not find colocalization with nuclei, cell membranes, or mitochondria.
- Fluorescence in various organic solvents:
The manuscript does not mention the need to measure fluorescence properties in different organic solvents. Evaluating the fluorescence characteristics of the sensor in different polar and non-polar organic solvents is useful to assess how the sensor performs in real samples and biological environments. In addition, solvent-dependent stability information can help select suitable solvents for long-term storage or use. Data on fluorescence properties in different solvents can deepen our understanding of the application range and behavior of the sensor.
We agree that the polarity of the environment can influence the fluorescence properties of the probe. Consequently, the absorption and emission spectra of GFZnP BIPY were recorded in a set of solvents with various polarities (MeOH, DMSO, THF, DCM, and toluene). The absorption spectra show differences in the molar absorption coefficient, but no significant shifts in the wavelength compared to the aqueous solution. No fluorescence was observed in either solvent. However, we extended our study to a viscous solvent, glycerol, in which we observed a redshift and significant fluorescence, similar to the ones observed in Zn2+ containing solutions. This result supports our hypothesis, that the non-radiative relaxation happens via the conformational movement of the free probe, which is attenuated in the viscous solvent. In real applications, such a high-viscosity environment is rarely present, eliminating the risk of interference. The results are now discussed in the manuscript and the spectra are shown in Fig. S14 of the supplementary information. Unfortunately, the timeframe of this revision does not allow us to carry out new stability studies in different solvents for long-term storage. However, we have already performed such a study in a mixture of DMSO and EtOH, which proved the stability of the sensor during 4 months. This result is now mentioned in the manuscript and thus, we recommend this solvent mixture for long-term storage.
Round 2
Reviewer 1 Report
Comments and Suggestions for Authors
Dear Authors,
I am satisfied with your response to my comments.
Reviewer 2 Report
Comments and Suggestions for Authors I am in agreement with the acceptance of this paper as the authors have adequately addressed the concerns of my peer review.